# Pulsed Spherical Tokamak—A New Approach to Fusion Reactors

**Mikhail Gryaznevich \*, Valery A. Chuyanov and Yuichi Takase** 

Tokamak Energy Ltd., 173 Brook Drive, Milton Park, Abingdon OX14 4SD, UK;
valery.chuyanov@tokamakenergy.co.uk (V.A.C.); yuichi.takase@tokamakenergy.co.uk (Y.T.)
\* Correspondence: Mikhail.gryaznevich@tokamakenergy.co.uk; Tel.: +44-7827914654

**Abstract:** Traditionally, spherical tokamak (ST) reactors are considered to operate in a steady state. This paper analyses the advantages of a pulsed ST reactor. The methodology developed for conventional tokamak (CT) reactors is used and it is shown that advantages of a pulsed operation are even more pronounced in an ST reactor because of its ability to operate at a higher beta, therefore achieving a higher bootstrap current fraction, which, together with a lower inductance, reduces requirements for magnetic flux from the central solenoid for the plasma current ramp-up and sustainment.

**Keywords:** spherical tokamak (ST); reactor; pulsed reactor

## 1. Introduction

The spherical tokamak (ST) plasmas, where the aspect ratio of the toroidal plasma $A = R_0/a$ is less than 2, have been predicted by A Sykes et al. to have several advantages over those with higher aspect ratios [1] because of the ability of STs to provide highly efficient use of the magnetic field to contain a plasma, although these predictions were met with scepticism [2]. However, a tokamak with 2.6 m major radius and aspect ratio of 1.9 had already been proposed by that time by D Jassby [3] to demonstrate thermonuclear ignition. Following this, after the ST path to fusion was advocated by Peng [4] and Stambaugh [5], experimental, computational, and theoretical studies have produced significant new results confirming the viability of this approach. The paper by Peng and Hicks [6] is one of the first that analysed the results of initial investigations into the engineering feasibility and some of the engineering issues specific to ST reactors, e.g., divertor power loads and the design of the replaceable central post, as well as advantages of STs in physics, e.g., the possibility of a high self-driven bootstrap current [7] that reduces requirements for expensive external current drive (CD) for steady-state operation. The conclusion has been drawn that the technical requirements are less demanding than those in conventional tokamak (CT) reactors, but the physics advantages need to be confirmed experimentally.

Studies by Stambaugh et al. [5] concluded that the high beta potential of the ST is so great that the physics of this device will not determine its size. According to [5], even a very small ST device (major radius ~1 m) can produce ~800 MW (thermal), 200 MW (net electric) power and would have a gain (gross electric power/recirculating power), $Q_{plant}$ of ~2. It was also concluded that the ST approach has two key features of a realizable commercialization strategy: a low-cost pilot plant that can attract commercial cost sharing at an affordable level and with minimal financial risk, and a strong economy of scale, leading to power plants that are still small on an absolute scale. Following this, detailed physics and engineering studies have been performed by the ARIES-ST team [8] and other ST reactors have been proposed [9,10].

Both ST and CT reactor studies usually assume that the duration of plasma burning in a reactor must be sufficiently long because the steady-state operation is likely to result in the least expensive tokamak reactor as the cost of electricity (CoE) strongly depends on the plant availability. However, to date, although a possibility of steady-state relevant operation

regimes has been demonstrated [11], there are no experimental results demonstrating steady-state plasmas with reactor relevant plasma parameters (only low-performance multi-hour discharges have been achieved), and one cannot expect such results to be achieved in the foreseeable future. There are no proper experimental facilities since devices for such studies are very expensive and risky. A big risk is that even if technology issues associated with steady-state magnets, heating and CD auxiliary systems, and requirements for their maintenance could be resolved, there is growing evidence that the duration of the burning pulse is limited by other factors. Even in relatively short pulses, unexplained fast disruptions have been observed, which can be attributed to dust generation, flaking of depositions on the vessel wall, formation of hot spots, and other plasma–wall interaction phenomena. Consequently, before discussing the availability of a Plant and CoE, one should provide solutions to resolve these issues (including maintenance solutions, e.g., dust removal). Pulsed fusion reactors may resolve some of these issues and have been considered in the past [12–15], and results of the most detailed and recent study are reported in [16].

An ST reactor can have a reasonably small fusion power and may be used as a basis for a modular fusion power plant [17]. The concept of a modular plant is a natural way to combine pulsed reactor modules in a continuously operating power plant with high availability of the entire plant, even if each module operates in a pulsed mode. As shown in [16] and in other studies [12], the minimum pulse duration to get to a sustained burn should not be less than 10 min, but a commercially attractive pulsed reactor should have a pulse duration of 1.5–2 h. Devices with burn duration of tens of minutes could be suitable for prototype studies. Although tens of minutes or even 2 h discharges are far from typically assumed steady-state durations, the limitations mentioned above may be serious showstoppers, and technical limitations, e.g., an increase in the size of the reactor to overcome the resistive losses incurred by a copper centre post (Stambaugh et al., vision based on the Peng–Hicks replaceable copper centre post concept) and low CD efficiency already constrain even a pulsed reactor design.

Now, thanks to a new technology—high-temperature superconductor (HTS) magnets— a different strategy to resolve some of these issues looks possible. Not only does the toroidal field (TF) magnet have several advantages when HTS is used (higher field, higher temperature of the coolant, higher acceptable stresses, compact size), but the use of HTS in the tokamak central solenoid (CS) can also provide an order of magnitude higher magnetic field and flux to drive the plasma current inductively. It is reasonable to start studies of reactor-relevant plasmas with relatively long, but not steady-state, pulses, and tens of minutes could be considered as a starting point. Eventually, these studies can lead to either a steady state or a longer-burn pulsed reactor. As shown in [16], a pulsed CT reactor with HTS CS can be economically acceptable.

The application of these results to STs is not straightforward. STs can provide a higher beta (ratio of the plasma pressure to the required magnetic pressure) and a higher bootstrap current fraction. Their energy confinement is less sensitive to the magnitude of the plasma current [18–21], so the plasma current is mainly needed to ensure the confinement of alpha particles, not to enhance the plasma performance.

The advantages of a pulsed ST reactor are analysed in this paper in a similar way as in [16] for a pulsed CT reactor. Plasma current ramp-up and sustainment in an ST are discussed in Section 2. Comparison of steady-state and pulsed options for an ST reactor is made in Section 3. Conclusions are given in Section 4.

## 2. Plasma Current Ramp-Up in a Pulsed ST Reactor

### 2.1. Previous Studies, Experiments, and Theory

The use of a CS for plasma current initiation and ramp-up, as in CTs and in some STs (MAST-U, NSTX, GLOBUS-M), may simplify the reactor design. However, using the solenoid flux for plasma formation and current ramp-up is a much less efficient use of the volt-second capability of the CS than to use it to sustain the plasma current at the flat top. At the flat top, this flux will be used to compensate the resistive losses, which are much

lower during the flat top than during the initial phase with lower electron temperature and higher resistivity. In addition, significant inductive flux needs to be supplied during current ramp-up from plasma start-up to the full current.

A solution to this problem using the $B_V$ ramp-up has been indicated in [22]. It was recognised that the external equilibrium coils can provide not only the vertical field needed for plasma equilibrium but also the poloidal flux to increase the plasma current when the plasma stored energy increases due to external heating and, in fusion reactors, due to the self-heating by alpha particles.

Such $B_V$ ramp-up has been successfully demonstrated on MAST [23,24] and on JT-60U [25]. Figure 1 presents an example of $B_V$ ramp-up from $I_p = 250$ kA to 500 kA at a constant loop voltage when the plasma was heated by 0.7 MW of NBI. Here, plasma current, loop voltage, plasma geometric axis, NBI power, vertical field coil current, and $\beta_{pol}$ traces are shown. However, this technique can be applied only to the plasma that is already formed and heated. The exact moment when the plasma heating should start and the vertical field is increased to balance the plasma expansion due to the increase in the plasma pressure is determined by optimising the $B_V$ ramp-up condition [22–24]. For the most efficient $B_V$ ramp-up, it should start at the highest possible $\beta_{pol}$ and the high enough electron temperature. On MAST, NBI of 2 MW was enough to produce $B_V$ ramp-up at zero loop voltage from the CS. Efficient $B_V$ ramp-up has been achieved at $\beta_{pol}\sim0.8$–1 and $T_e\sim1$–1.5 keV [24].

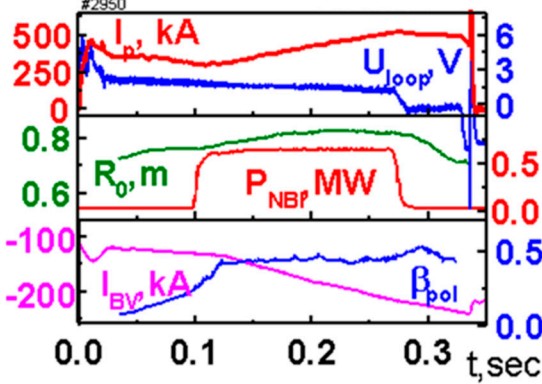

**Figure 1.** Plasma current ramp-up by $B_v$ ramp-up on MAST [23] at constant solenoid applied loop voltage: $I_p$-plasma current, $U_{loop}$-loop voltage, $R_0$-plasma geometric axis, $P_{NBI}$-NBI power, $I_{BV}$-vertical field coil current, and $\beta_{pol}$-poloidal beta.

The $B_V$ ramp-up increases the plasma current due to two mechanisms. The first mechanism has the plasma heating on a characteristic timescale. Magnetic surfaces move outwards due to the Shafranov shift, which results in the plasma current increase on each flux surface simultaneously. The second mechanism is due to the poloidal flux from the changing $B_v$, which contributes to the external loop voltage, and the plasma current increases on the resistive timescale.

The simplest model that includes the effect of the $B_V$ ramp-up is described by the poloidal flux $\psi$ evolution equation [24]:

$$\sigma(T_e, n)\frac{\partial \psi}{\partial t} = \frac{J^2 R_0}{\mu_0 \rho}\frac{\partial}{\partial \rho}\left(\frac{G_2}{J}\frac{\partial \psi}{\partial \rho}\right) - \frac{V'}{2\pi \rho}j_{ext} \tag{1}$$

Here, $\sigma$ is the plasma conductivity, $\rho$ is the plasma minor radius normalised by the toroidal flux, $V$ is the volume inside a flux surface, $V' = dV/d\rho$, $G_2$ is the metric coefficient, $G_2 = V'/4\pi^2 <(\nabla\rho/r)^2>$, $J$ is the poloidal current, and $j_{ext}$ is the current density, including

the bootstrap and other externally driven currents (but excluding the contribution from $B_V$). The plasma current inside the surface $\rho$ is expressed by the formula

$$I_p(\rho) = \frac{G_2}{\mu_0} \frac{\partial \psi}{\partial \rho} \qquad (2)$$

And can increase due to the increase in $G_2$, keeping the poloidal and toroidal fluxes constant (inductive mechanism). The effect of an additional flux through the plasma can be included in the boundary condition for Equation (1):

$$d\psi/dt \mid_{edge} = - U^{Bv} - U^{CS} - U^{pl} \qquad (3)$$

where contributions from $B_V$, the central solenoid (CS), and the plasma current evolutions are summed.

The plasma pressure increases due to heating or a density rise changes the equilibrium condition by increasing the $G_2$ factor. For a high enough $\beta_{pol}$, this contribution becomes higher than the resistive losses and the plasma current stays fixed with low external loop voltage ($d\psi/dt \mid_{edge} = -U^{Bv}$), or even increases. The latter corresponds to the "overdrive condition".

However, the equilibrium field must be constant during the steady-state burning phase and cannot be used to provide volt-seconds to keep the plasma current constant against resistive dissipation. In [22], it was proposed to have a CS as an efficient tool for current drive, but with a very small flux swing. This solenoid could be used to trim the plasma current in case of external disturbances, resistive decay, or to limit the plasma current at a desired level if it goes too high. The new strategy of the development does not require non-inductive CD, except possibly for the initial start-up, and for solenoid recharging [26] if it is needed to use the solenoid repeatedly during the burning phase, in the case that resistive losses require such recharging to prolong the plasma pulse to a desired duration and the volt-second capability of the engineeringly feasible CS is not sufficient. It will be shown below that the new technology (HTS) permits an increase in the flux of the CS in ST by an order of magnitude. All these factors allow the HTS CS to be considered as a viable tool to provide long pulses of fusion burn in STs.

### 2.2. Possible Scenario and the Main Requirements

The plasma should first be formed and heated to the required temperatures. Different non-inductive techniques have been proposed and demonstrated for start-up and ramp-up [25,27,28], in which case the volt-seconds from the CS are not needed for plasma current ramp-up and can be saved for sustaining the flat-top. The high electron temperature produced during start-up and by the applied core auxiliary heating will contribute to an increase in the pressure gradient and so to the bootstrap current $I_{bs}$ [7], opening a possibility for the bootstrap current to supplement the $B_V$ ramp-up and even the bootstrap overdrive. The bootstrap overdrive can be achieved in an ST due to a high bootstrap current fraction achievable in STs [8,29]. Simulations with the DINA/ASTRA 1D evolution code have been performed for the current ramp-up phase of the STEP device (R = 1.2 m, a = 0.75 m, $B_t$ = 3.5 T, $I_p$ = 5 MA, $\kappa$ = 3, $\delta$ = 0.4), as proposed in [30]. Conditions have been optimized for the high bootstrap fraction $f_{bs}$. Lower values for the target plasma current (3MA) and toroidal field (2T), $P_{NBI}$ = 10 MW, $T_{e,i}(0)$~3.0 keV, $n_e(0)$~$1.7 \times 10^{20}$ m$^{-3}$, have been chosen for these simulations to bring parameters close to those achieved on MAST [23]. Figure 2 presents the evolutions of the total plasma current (dashed line) and $I_{bs}$ (solid). The analysis shows that the bootstrap overdrive can be achieved even for these modest parameters.

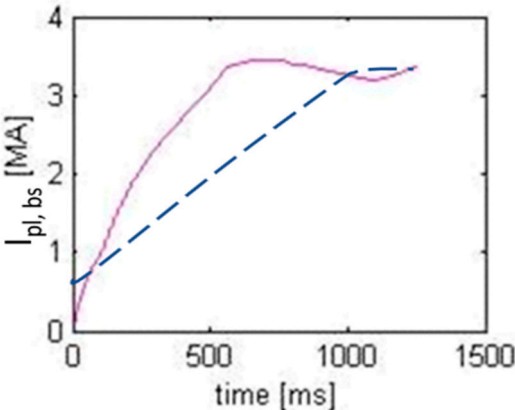

**Figure 2.** Evolutions of the total plasma current (dushed line) and the bootstrap current $I_{bs}$ (solid), calculated by DINA-ASTRA simulation for STEP [30]. The bootstrap overdrive condition is achieved in this example.

A possible time sequence of operation of a pulsed ST reactor is shown conceptually in Figure 3. Plasma break-down is assisted by the ECR wave at a very moderate power of tens of kW. When the closed flux surface configuration is formed, an efficient ECR/EBW current drive can be expected [25] to increase the plasma current to a MA level using a few MWs of microwave power. This phase may take from a hundred to several hundred ms and will be efficient at a low plasma density, around $1–2 \times 10^{19}$ m$^{-3}$. At this low density, the plasma temperature should reach values required for an efficient $B_V$ ramp-up. At this stage the plasma density is increased and ICRH power is added. Such heating should produce quite peaked temperature profiles. Supplemented by the core fuelling by pellet injection, peaked pressure profiles are expected, providing a high bootstrap current, which will contribute to the $I_P$ ramp-up. The density will be increased further to the level needed to achieve fusion burn. At such high densities, the efficiency of ECCD will be reduced (dashed $I_{EC}$ curve), however the efficiency of RF heating and $B_V$ ramp-up will not. For the ramp-up to the full $I_P$, the use of the CS flux is not assumed.

Although the bootstrap current in an ST is expected to contribute up to high levels of ~90% of the total plasma current at the flat-top, the remainder (~10%) must be driven by other means. The required externally driven current may be even less if the bootstrap fraction can be increased to 95% or more, as suggested in [8]. The use of steady-state RF CD is not efficient as it will reduce the $Q_{plant}$ value, which is not acceptable as at this stage the energy production will start and can be utilised. The remainder of the current should be driven inductively using the CS. Given that the CS is used only for the $I_P$ maintenance and not for the $I_P$ ramp-up, the needed CS volt-seconds can be much smaller than the full CS volt-seconds that would be needed for the $I_P$ ramp-up at the initial stage. When the limited CS volt-second capability is exhausted before the end of the burning stage, it will need to be recharged. Both microwaves and RF can be applied at this stage. RF and microwave auxiliary systems that were used during the ramp-up phase can be re-applied for recharging. This will reduce the efficiency of the energy production because of the increased recirculating power, but only for a considerably short period. More detailed analysis is needed to find out the maximum duration of the burning phase before recharging is needed. If the bootstrap fraction can be increased, only a few (or no) recharging would be needed. When recharged, the CS can be used again, as shown in Figure 3.

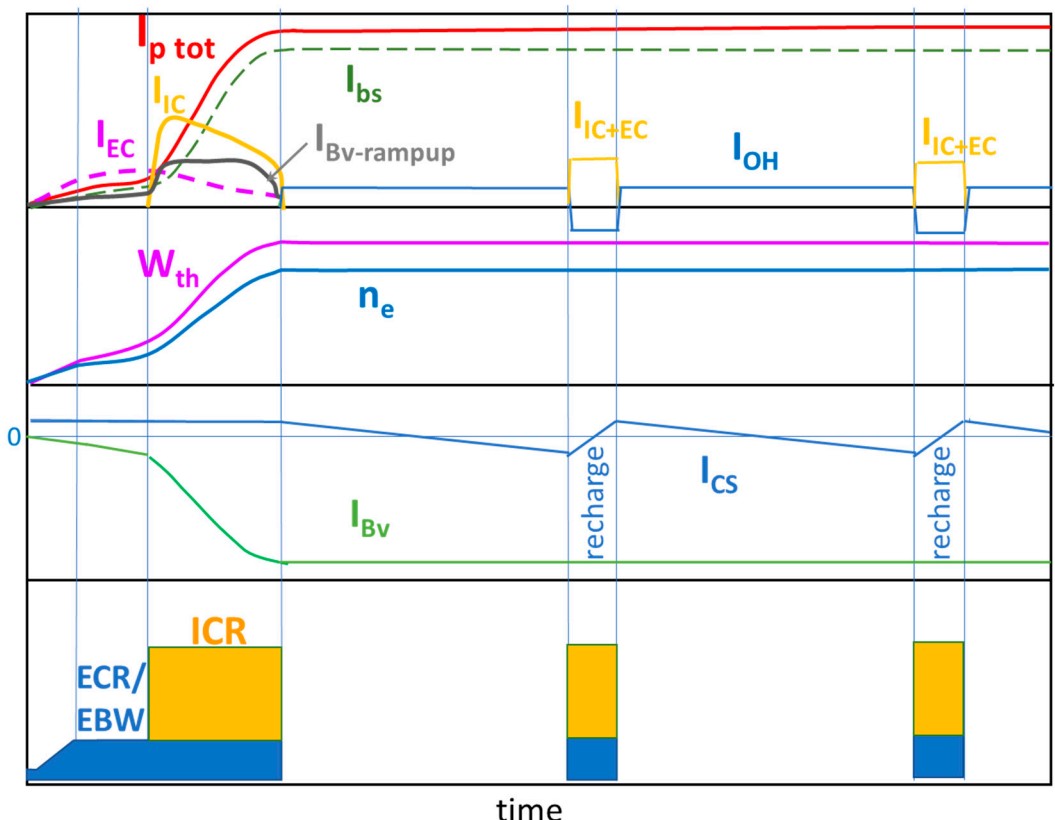

**Figure 3.** Proposed time sequence of the operation: $I_{p\,tot}$-total plasma current and its components $I_{EC}$, $I_{IC}$, $I_{bs}$, and $I_{Bv\text{-}rampup}$ (not to scale); $W_{th}$-plasma thermal energy and $n_e$-electron density; $I_{CS}$-CS current and $I_{BV}$-vertical field coil current; and auxiliary CD and heating power application periods.

The flux that the CS needs to supply at the flat-top during the burning phase must compensate the resistive loss. Estimates show that for a typical ST pilot plant scale device (this device is referred to as ST280-5T in this paper) with $R_0$ = 2.8 m, A = 1.9, $B_t$ = 5T, $I_p$ = 10 MA, a loop voltage of about 1–2 mV is needed to sustain the $I_p$ flat-top. This means that the CS with 2 Vs can sustain the $I_p$ flat-top for 1000–2000 s. As will be shown below, incorporation of such a CS in the radial build of an ST reactor would not be an issue. Assuming a bootstrap fraction of 90%, the RF power required for recharging is about 20 MW (depending on the duration of recharging), but this estimate will need experimental confirmation on a high-field burning plasma ST.

To have the longest pulse, one needs to operate with the highest bootstrap fraction. This requires a high $\beta_{pol}$, which is constrained by the high enough value of the plasma current to confine alpha particles during the burning phase and on the way to burning. For example, it was shown [31] that even for a small ST with $R_0/a$ = 0.6/0.4 m, $B_t$ = 5 T, $\kappa$ = 3, $n_e$ = 1.2 × 10$^{21}$ m$^{-3}$, $I_p$ = 5–6 MA is needed to confine 80% of alpha particles. The compromise must be carefully studied and depends on the dimensions of a reactor (which constrains the size of the CS) and on requirements of the pulse duration. This analysis is important for a commercial reactor, but it is not needed for our first goal—to get to a sustained burn in a commercially attractive pulsed reactor with a pulse duration of 1.5–2 h [12,16] or longer [13–15].

## 3. Comparison of the Pulsed and Steady-State ST Reactor Options

As there is a pulsed operation option for an ST reactor without relying on expensive and low-efficiency RF or microwave CD, the results of [16] can be now used, taking into account the advantages of the ST approach—more efficient $B_V$ ramp-up, higher bootstrap current, and lower total inductance. The same primary and secondary cost metrics (the

toroidal magnetic field energy within the magnet volume per watt and the plasma volume per watt) are used here as in [16]:

$$C_{\text{MAG}} = W_{\text{TF}}/P_{\text{F}} \text{ MJ/MW} \quad \text{primary cost metric}$$
$$P_{\text{VOL}} = P_{\text{F}}/V_{\text{P}} \text{ MW/m}^3 \quad \text{secondary inverse cost metric} \tag{4}$$

where $W_{\text{TF}}$ is the toroidal magnetic field energy, $V_{\text{P}}$ is the plasma volume, and $P_{\text{F}}$ is the fusion power. A comparison of $C_{\text{MAG}}$ for ST and CT pulsed reactors producing the same thermal fusion power $P_{\text{F}}$ will show the relative attractiveness of each option.

For the ST280-5T device, $P_{\text{F}}$ = 800 MW and $W_{\text{TF}}$ = 11 GJ, and $V_{\text{P}}$ = 280 m³, which gives $C_{\text{MAG}}$ = 14 MJ/MW and $P_{\text{VOL}}$ = 2.9 MW/m³. This can be compared with $C_{\text{MAG}}$ of 33–37 and $P_{\text{VOL}}$ of 2.1–3.8 for pulsed ARC, depending on the operating density [16]. While $P_{\text{VOL}}$ is comparable to ARC, $C_{\text{MAG}}$ is more than a factor of two smaller for ST280-5T, indicating the advantage of ST. A conclusion from the comparison of pulsed and steady-state reactors for a CT in [16] is that a steady-state reactor needs about 50% higher $Q_{\text{fus}}$ than a pulsed reactor because of its lower RF CD efficiency, unless the confinement enhancement factor H (which is the ratio of the confinement time to that predicted by the ITER98(y,2) scaling in [32]) is increased. For a pulsed reactor, the $Q_{\text{fus}}$ constraint does not play such a major role since no or little RF and microwave CD is required and the current at the flat-top is driven, fully or nearly fully, by the CS. The high bootstrap fraction in an ST, more efficient $B_V$ ramp-up, and potentially better confinement (at high TF, see [18–21]) are also beneficial.

As concluded in [16], for a steady-state reactor with the RF or microwave CD, the CS demands are much smaller than for a pulsed reactor—sufficient volt-seconds are required only to raise the plasma current to its final desired operating value. This is a small fraction of the total requirement in a pulsed reactor. As discussed above, operation of a pulsed ST reactor assumes that the start-up and the current ramp-up can be accomplished without the use of the CS. This makes a big difference and the statement that a pulsed reactor requires more solenoid volt-seconds should be revised for an ST. As was shown above, the ST approach makes these more efficient than in a CT case, which allows for a reduction in the requirement of the CS, only to sustain the plasma current at the flat-top. If the engineering of the CS will still significantly affect the size of the device, recharging of the CS during the flat-top may also be considered in order to reduce the CS size.

The HTS CS can provide the flux swing sufficient for our goals. In contrast to [16], for the CS in ST operation at the maximum possible magnetic field for the HTS magnet is not required. There are two reasons to limit the optimum magnetic field—mechanical stresses and limited current density in the CS winding pack.

A mechanical structure that can support the CS decreases the available cross-section of the winding pack for the fixed radius of the central post. Following [16] and assuming the maximum permissible stress in the structure $\sigma_{\text{max}}$ ~500 MPa, the ratio of thickness of the mechanical structure $\delta_{\text{cs}}$ to the external radius of the winding pack $R_{\text{cs}}$ may be estimated as

$$\delta_{\text{cs}}/R_{\text{cs}} = B_{\text{cs}}^2/(2 \mu_0 \sigma_{\text{max}}) = 0.32 (B_{\text{cs}}/20)^2 \tag{5}$$

where $B_{\text{cs}}$ is the peak field of the CS (in Tesla). The final thickness of winding pack ($\delta_{\text{wp}}$) decreases the mean magnetic field and magnetic flux in the bore of the CS. The thickness of the winding pack is determined by the achievable mean current density $J_{\text{wp}}$ and by the selected maximum field

$$\delta_{\text{wp}} = B_{\text{cs}}/\mu_0 J_{\text{wp}} \tag{6}$$

The typical values of $J_{\text{wp}}$ are ~80 MA/m² and may be increased after some development up to 100 MA/m² [33].

Taking both effects into account, it was found that for the most economical CS, the field $B_{\text{CS opt}}$ depends on the current density in the winding pack and the external radius of the CS $R_{\text{ext}} = R_{\text{CS opt}} + \delta_{\text{CS}}$ (see Figure 4) and the optimum field at the solenoid is in the range 8–15 T compared to 23 T found in [12] for a CT reactor.

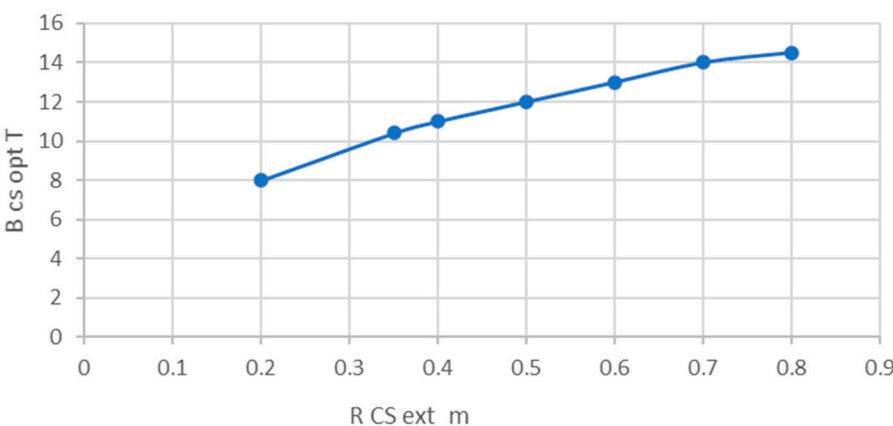

**Figure 4.** Optimum maximum CS field $B_{cs\,opt}$ as a function of the external radius of CS, $R_{CS\,ext}$, calculated for $\sigma_{max}$ = 500 MPa and a constant current density in the winding pack $J_{wp}$ = 80 MA/m$^2$.

The radius of the CS in an ST that satisfies these constraints, depending on the required volt-seconds, is shown in Figure 5. The calculated dependence of magnetic flux on the external radius of the central solenoid $R_{CS\,ext}$ is well approximated by $\psi(R_{CS\,ext})$ = 8.6 2$\pi$ $R_{ext}$ $^{2.5}$ [Vs]. As discussed above, to ramp up the plasma current in an ST reactor, about 20 Vsec may be required, and the radius of the CS should be at least 0.7 m. However, if one uses the CS only to sustain the plasma current at the flat-top, the radius can be as small as 0.2–0.3 m, which may be accommodated even for a CS positioned on the inboard side (inside) of the TF magnet central post [33]. In this case, the stresses on the HTS CS can be even more reduced as the field from the TF magnet will be much smaller.

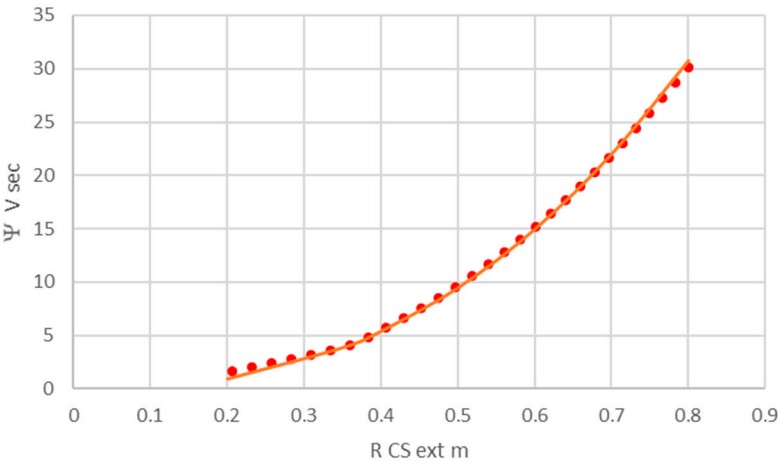

**Figure 5.** Magnetic flux of the CS $\psi$ at the optimum field as function of the external radius of the CS, $R_{CS\,ext}$, calculated for current density in the winding pack $J_{wp}$ = 80 MA/m$^2$, and $\sigma_{max}$ = 500 MPa.

Another conclusion of the comparison of pulsed and steady-state CT reactors [16] is that the steady-state reactor requires a much higher value of the safety factor at the axis q(0) (4.75 vs. 1 for a pulsed option). It is known [34] that such an increase in q(0) in CTs results in a significant decrease in the ideal no-wall $\beta_N$ limit and, consequently, a reduction in the bootstrap current fraction. Additionally, regimes with such high q(0) have not been evaluated and demonstrated experimentally for monotonic q-profiles. In an ST, regimes with a monotonic q-profile can be achieved even for a non-monotonic plasma current density profile.

It was also concluded in [16] that the reduction of the aspect ratio in a steady-state reactor requires some increase in the H-factor, about 25% when going from A = 5 to A = 2. Such an increase can be confidently expected in STs. In a pulsed reactor, this requirement is even softer. Such an analysis has not been done in detail in [16], but good confinement

in STs may result in a lower $P_F$, which is needed to sustain burning. The economics of a power plant will not suffer from such a reduction due to the possibility of a modular approach. A desirability for a CT steady-state reactor to increase the output power may result in a design driven more by technology than by plasma physics [35]. Problems of the neutron wall loading and heat flux to the divertor will be substantially worse. An increase in the reactor size will also result in an increase of the required auxiliary CD systems, which may complicate finding a supply chain for such systems (e.g., gyrotrons). Additionally, as mentioned in [16], very large plants, either pulsed or steady-state, may not be desirable by the industry because of grid concerns, large capital investments, long construction times, logistics, etc. These arguments are valid both for ST and CT reactors.

As was mentioned above, the use of the HTS in the TF magnet can allow a significant increase in the toroidal field compared to the low-temperature superconductor TF magnet. Analysis for a CT pulsed reactor has shown that the major radius $R_0$ decreases and the corresponding power density $P_{VOL}$ increases as the TF magnetic field ($B_t$) increases. In an ST, the required increase in confinement and increase in the cost $C_{MAG}$ and a conclusion of the beneficial reduction in TF are softened by a stronger dependence of the confinement on $B_t$ and, consequently, lower requirements for the increase in $B_t$. It is much easier to simultaneously satisfy the Troyon $\beta$ limit and the kink q limit in an ST. The conclusion in [16] is that the 'best' design makes use of the lowest possible $B_t$. This favours the ST geometry, and even in the case of CT, the required $B_t$ can be reduced by a factor of 1.5 in a pulsed reactor compared to a steady-state reactor. Therefore, the conclusion of [16] that a pulsed reactors are competitive and more desirable than steady-state reactors is even more emphasised for an ST reactor.

## 4. Conclusions

The analysis described in [16] has been extended to tokamaks with low aspect ratio, STs. It can be concluded that the advantages of a pulsed reactor include:

- Reduced need for very expensive non-inductive current drive;
- Lower risk as the steady state operations may have limitations;
- Broader range of possible plasma parameters (not limited by the CD requirements);
- Simpler, faster, and cheaper development path as operations with the CS are well established;

These are even more pronounced than the high aspect ratio case due to several specific features of the ST:

- Possibility to operate at a higher beta;
- Possibility to achieve a higher bootstrap current fraction;
- Stronger increase in the confinement with the toroidal field;
- Better stability, both for -pressure-driven micro instabilities and for the current-driven instabilities.
- Lower requirements on the volt-second capability of the CS due to the ability to reach full plasma current without using the CS.

Some of these features have already been demonstrated on STs. However, more experiments and new experimental facilities are needed to confirm the advantages of the ST for a fusion reactor. Another important conclusion of these studies is that as the CS provides the most efficient method of suppressing the plasma current variation during the flat-top, it should not be removed completely from an ST reactor if a constant plasma current operation is desired.

**Author Contributions:** Conceptualization, all authors; writing—original draft preparation, V.A.C.; writing—review and editing, M.G. and Y.T. All authors have read and agreed to the published version of the manuscript.

**Funding:** This research received no external funding.

**Acknowledgments:** We acknowledge A. Dnestrovskii, R. Slade, and I. Voitsekhovitch for useful comments.

**Conflicts of Interest:** The authors declare no conflict of interest. The funders had no role in the design of the study; in the collection, analyses, or interpretation of data; in the writing of the manuscript; or in the decision to publish the results.

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
