# Peer review of "Pulsed Spherical Tokamak—A New Approach to Fusion Reactors"

_plasma, doi:10.3390/plasma5020019_

Round 1

Reviewer 1 Report

This manuscript requires improvement before it can be considered for publication.

Improve English grammar and paper structure. (should not have consecutive sentences starting with the same word, avoid the use of we, poor word choice, improve sentence structure, no one sentence paragraphs, proper use of caps is necessary, definition of terms and variables, etc)

--avoid lumping references...discuss each individually describe the lines. 

--it is unclear what figures and results are new and what are previously reported. Please clarify

--captions should be more descriptive and include legends that

--present references in order

--add current references (less than 5 yrs old)  there are only 4/36 current references in this manuscript. Additionally, add MDPI references

--improve the conclusions...they need to be more detailed and better supported

--section titles should be improved. Current titles are poor and not classic types.

--include accuracy and sensitivity of results.

--equation number for each equation is necessary

Author Response

Referee 1. This manuscript requires improvement before it can be considered for publication.

Thank you very much for constructive recommendation. The abstract and conclusions are amended following recommendations.

- Improve English grammar and paper structure. (should not have consecutive sentences starting with the same word, avoid the use of we, poor word choice, improve sentence structure, no one sentence paragraphs, proper use of caps is necessary, definition of terms and variables, etc) => several changed are made

--avoid lumping references...discuss each individually describe the lines. => done

--it is unclear what figures and results are new and what are previously reported. Please clarify => all figures are either new or re-drawn for this manuscript. References to results from other studies are now given

--captions should be more descriptive and include legends that => comment not comleted, not clear

--present references in order => done

--add current references (less than 5 yrs old)  there are only 4/36 current references in this manuscript. Additionally, add MDPI references => references follow MDPI requirements

--improve the conclusions...they need to be more detailed and better supported => done

--section titles should be improved. Current titles are poor and not classic types. => Don’t understand what “classic” means

--include accuracy and sensitivity of results. => this is not applicable for conceptual studies as no analysis of experimental data is given

--equation number for each equation is necessary => corrected

Reviewer 2 Report

After a long period of gas discharges that evolved into a seeming preference for the conventional tokamak design for fusion plasmas, stellarators and spherical tokamaks have become of interest again in a search for designs that may circumvent or overcome problems of the conventional design. The once claimed superiority of the classical tokamak design may not be that pronounced after all, and questions of size (cost) and operation mode (long pulse versus continuous) have resurfaced in the quest for practicality. The present authors present a 
particular scheme in which a pulsed mode of a spherical tokamak is investigated. No clear favourite has emerged from the competition yet, and it is still possible that such a variant may emerge with promising features, even before running ITER and preparing DEMO might drain all the available research funds. 

The present study is based on simulations of a demonstration device that take input from current research results. This procedure is obviously cheaper and faster than building a device first. Of course, sooner or later some reality checks will be needed for validating any such simulation, as is obviously clear to the authors.  

They write in clear English, and they are not only fluent in that language, but in the scientific and technical topics of the field as well. The discussion follows that of a recent exercise for a cw reactor design and tries to produce corresponding parameters and costs for a pulsed one. A major cost reduction factor is the use of HTS coils. 
Any pulsed reactor, advantageous as it may seem, has to be thought of as several modules that form a power plant with a continuous output. Hence cost arguments will have to reach farther than the performance of a single module. However, the considerations discussed and the trends indicated by the simulations are interesting and worth publishing in MDPI Plasma. 

line 297, "tor" wants to read "for" 

Author Response

Referee 2. Thank you for very positive review!

line 297, "tor" wants to read "for" => corrected

Round 2

Reviewer 1 Report

Some modifications have been made in this manuscript, however more is required.

This manuscript requires improvement before it can be considered for publication.

- Improve English grammar and paper structure. 

--captions (figure descriptions) should be more descriptive and include legends that describe the data. Each figure needs to be discussed

--add more current references (less than 5 yrs old)  there were only 4/36 current references in the original manuscript. More current references are required

--section titles should be improved. Current titles are poor  . Use tites such as Results, theory, analysis. Then use subtitles if necessary

--include accuracy and sensitivity of results based upon the assumptions that are made

Author Response

Response to Referee 2

“Some modifications have been made in this manuscript, however more is required.

This manuscript requires improvement before it can be considered for publication.”

- Improve English grammar and paper structure.

=> we can’t find where the grammar could be improved, please specify

 --captions (figure descriptions) should be more descriptive and include legends that describe the data. Each figure needs to be discussed

=> captions include descriptions, e.g. plasma current, loop voltage, plasma geometric axis, NBI power, vertical field coil current and bpol. for Fig.1

Typically, captions should only explain what is plotted, discussions are in the text. We’ve made several improvements

--add more current references (less than 5 yrs old) there were only 4/36 current references in the original manuscript. More current references are required

=> unfortunately we can’t find any new relevant references. Please suggest.

--section titles should be improved. Current titles are poor. Use titles such as Results, theory, analysis. Then use subtitles if necessary

=> titles such as Results, theory, analysis would not be easy to use in a conceptual paper. We’ve made several changes.

--include accuracy and sensitivity of results based upon the assumptions that are made

=> we are not discussing experimental results and measurements, so this is not applicable. As stated in the Conclusion, more experiments and new experimental facilities are needed to confirm the advantages of the ST for a fusion reactor described in this paper.

Round 3

Reviewer 1 Report

Little has changed from the previous revision.

This manuscript requires improvement before it can be considered for publication.”

  • Improve English grammar and paper structure. (i.e improve word choice and sentence structure throughout, excessive number of sentences starting with the same word, )
  •  

--captions (figure descriptions) should be more descriptive and include descriptions of the legends .  A figure should be able to exist independent of the manuscript - data should be clear and the description should not use only variables but descriptions 

--section titles should be improved. Current titles are poor  . Use titles such as : Results, theory, analysis. Then use subtitles if necessary.. You still have an analysis section an a theory section - your reasoning not to use such titles is not convincing.

--include accuracy and sensitivity of results based upon the assumptions that are made. Your assumptions to simplify the problem need to  be better discussed and quantified. --your reasoning not to use such titles is not convincing.

Author Response

This manuscript requires improvement before it can be considered for publication.”

  • Improve English grammar and paper structure. (i.e improve word choice and sentence structure throughout, excessive number of sentences starting with the same word, 

The authors have published overall more than ten hundred papers, so far we've never had such request.  Please give us specific, not general, recommendations. We think that the paper structure is adequate. We gave the paper to the former Chief Editor of Nuclear Fusion and he found it up to usual standards. For sure, there still may be some typos but they are not crucial for understanding the scientific content.

--captions (figure descriptions) should be more descriptive and include descriptions of the legends .  A figure should be able to exist independent of the manuscript - data should be clear and the description should not use only variables but descriptions 

We disagree with this. Figure captions should describe what is shown. Figures must be taken in the concept of the paper, but not independent.

--section titles should be improved. Current titles are poor  . Use titles such as : Results, theory, analysis. Then use subtitles if necessary.. You still have an analysis section an a theory section - your reasoning not to use such titles is not convincing.

We think that current titles are adequate. In a conceptual paper titles such as "results, theory, analysis" are not suitable. We also don't have "a theory section" - we are not presenting any new theories, just equations that explain some used definitions and straightforward analytic formulas that we use. The paper in not a theoretical one, neither an experimental, so we don't give analysis of new theories or new experimental data.

--include accuracy and sensitivity of results based upon the assumptions that are made. Your assumptions to simplify the problem need to  be better discussed and quantified. --your reasoning not to use such titles is not convincing.

Please be more specific. If error bars are really needed, please say where. General comment is not helpful.

Round 4

Reviewer 1 Report

There has been no change  to the manuscript to address concerns of the previous version. This anuscript still requires improvement.